# Cramér–Rao Lower Bounds on 3D Position and Orientation Estimation in Distributed Ranging Systems

**Sharanya Srinivas** , **Samuel Welker, Andrew Herschfelt * and Daniel W. Bliss ***

Center for Wireless Information Systems and Computational Architectures (WISCA), Arizona State University, Tempe, AZ 85281, USA
* Correspondence: andrew.herschfelt@asu.edu (A.H.); d.w.bliss@asu.edu (D.W.B.)

**Abstract:** As radio frequency (RF) hardware continues to improve, many technologies that were traditionally impractical have suddenly become viable alternatives to legacy systems. Two-way ranging (TWR) is often considered a poor positioning solution for airborne and other vehicular navigation systems due to its low precision, poor angular resolution, and precise timing requirements. With the advent of modern RF hardware and advanced processing techniques, however, modern studies have experimentally demonstrated TWR systems with an unprecedented, sub-centimeter ranging precision with low size, weight, power, and cost (SWaP-C) consumer-grade hardware. This technique enables a new class of positioning, navigation, and timing (PNT) capabilities for urban and commercial aircraft but also instigates new system design challenges such as antenna placement, installation of new electronics, and design of supporting infrastructure. To inform these aircraft design decisions, we derive 2D and 3D Cramér–Rao lower bounds (CRLBs) on position and orientation estimation in a multi-antenna TWR system. We specifically formulate these bounds as a function of the number of antennas, platform geometry, and geometric dilution of precision (GDoP) to inform aircraft design decisions under different mission requirements. We simulate the performance of several classic position and orientation estimators in this context to validate these bounds and to graphically depict the expected performance with respect to these design considerations. To improve the accessibility of these highly theoretical results, we also present a simplified discussion of how these bounds may be applied to common airborne applications and suggest best practices for using them to inform aircraft design decisions.

**Keywords:** uncertainty quantification; reliability analysis; localization; geometric dilution of precision (GDoP); Cramér–Rao lower bound (CRLB); two-way ranging (TWR); distributed coherence; unmanned aerial vehicles (UAVs); urban air mobility (UAM)

## 1. Introduction

As we continue to develop high-performance radio technologies, aerospace vehicles have become increasingly capable and versatile; unfortunately, the design of these platforms is a complex, multidisciplinary task rife with numerous design challenges [1]. As a result, many technologies that have demonstrated widespread success in terrestrial applications are not always suitable for airborne applications without careful consideration and redesign [2]. To successfully integrate a novel technology into modern aircraft, we must thoroughly understand both its own behavior and its interactions with the rest of the platform [3].

Positioning, navigation, and timing (PNT) are some of the most critical services for modern aircraft and numerous technologies provide these services through different mechanisms [4]. Two-way ranging (TWR) is traditionally considered a poor positioning solution for aerospace systems, but emerging technologies [5] have demonstrated sub-centimeter ranging performance with minimal size, weight, power, and cost (SWaP-C). These modern solutions not only enable a new class of distributed applications for urban

and commercial aircraft but also introduce significant system design challenges such as antenna placement [6], installation of new electronics [7], and the design of supporting infrastructure [8].

To inform these aircraft design decisions, we derive novel 2D and 3D Cramér–Rao lower bounds (CRLBs) on position and orientation estimation in a multi-antenna TWR system. These bounds are both a closed-form and tractable and are specifically formulated as a function of the number of antennas, platform geometry, and geometric dilution of precision (GDoP) to inform design decisions under different mission requirements. These bounds directly relate the position and orientation estimation precision to the geometric distribution of antennas on the platform. This can be used to infer the expected performance of existing installations or to inform the required distribution for new installations to achieve an arbitrary precision target. Since these geometries are often summarized by a single GDoP value, we also trace the performance manifold between ranging precision, positioning precision, and GDoP to graphically summarize the bounds, which can be explored visually without performing any calculations. To improve the accessibility of these highly theoretical results, we also present simplified commentary on how to navigate the trade space between performance and practical design considerations. These results are summarized into simple "best practices" to provide intuitive, geometric interpretations of the bounds. An example multi-antenna TWR configuration to which these bounds apply is depicted in Figure 1.

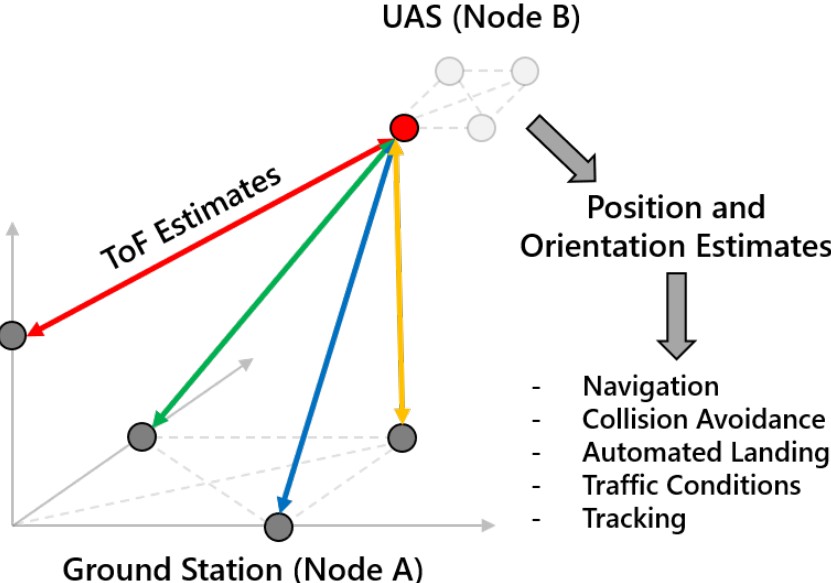

**Figure 1.** Example multi-antenna two-way ranging (TWR) system with a 4-antenna ground user *A* and 4-antenna aerial user *B*. By measuring the distance between each antenna pair (4 of which are depicted above using different colors to denote unique links), we can estimate the relative position and orientation of each user, enabling a wide variety of distributed airborne applications. We derive bounds on the performance of these estimators as a function of the number of antennas, platform geometry, and geometric dilution of precision (GDoP).

### 1.1. Background

The proposed Cramér–Rao lower bounds (CRLBs) on position and orientation estimation apply to estimators that use distance estimates as the primary measurement mechanism. This work was originally motivated by emerging TWR technologies for urban air mobility (UAM) applications, but the results apply to any technique that generates multiple range estimates for a given target. These results apply to several different types of TWR systems and are primarily enabled by previous work in the field of time-of-arrival (ToA) estimation.

### 1.1.1. Two-Way Ranging

One-way ranging systems such as radars [9] typically measure unresponsive or otherwise-uncooperative targets. Numerous extensions have been developed to refine the performance of this class of systems [10], but the uncooperative nature of the target imposes fundamental limits on the achievable precision [11]. Unlike these systems, two-way ranging (TWR) refers to a class of systems in which both the observer and the target communicate and cooperate to produce ranging estimates [12]. When these technologies were first emerging, radio-frequency (RF) hardware was large, heavy, and expensive, so static, robust, and reliable infrastructure such as a radar installation was a popular solution to the ranging problem [13]. As this hardware becomes more powerful and continues to shrink, however, installing modern TWR electronics on aircraft is significantly more viable [14].

As low-cost RF hardware became more accessible in the early 2000s, double-sided, two-way ranging became a somewhat more popular research topic [15]. Early theoretical work [16] demonstrated several tractable approaches assuming sufficiently behaved oscillators, closely followed by initial experimental demonstrations [17]. Several algorithmic extensions have since been explored [18] as well as more robust experimental demonstrations [19].

A fair amount of this early work was focused on traditional RF frequency bands, but the falling cost of higher-frequency equipment has motivated ultra-wideband (UWB) and even optical extensions. Since the precision of these techniques is proportional to the inverse of the signal bandwidth [20], high-frequency extensions can enable significantly better performance. Some studies have developed direct UWB extensions of the previous results [21], while others have explored the implications of integrating these high-frequency techniques with existing passive infrastructure [22]. While these high-frequency extensions enable intrinsically better performance, they also incur additional calibration challenges, which were characterized in [23].

In parallel, optical two-way ranging techniques have also been explored as high-precision ranging solutions [24]. These high-frequency systems have even better-ranging resolution than RF and UWB systems, but they also tend to have a more limited range and field of view, so they are suitable for different types of applications [25]. Recent studies have explored sophisticated 2D [26] and 3D [27] array processing extensions to further improve this performance. A comprehensive theoretical analysis for this class of systems was presented in [28], and modern studies continue to improve the fidelity of these techniques in real-world demonstrations [29].

### 1.1.2. Time-of-Arrival Estimation

For most TWR systems, time-of-arrival (ToA) estimation is the primary measurement mechanism that drives the rest of the signal processing chain. Naturally, the precision of these measurements will dictate the overall ranging precision of the system and is the subject of numerous studies in different applications. In indoor environments, researchers have characterized the performance of both wideband [30] and ultra-wideband [31] ToA estimators in dense multipath channels. Similarly for outdoor environments, several ToA estimation techniques were explored in [32,33], whose corresponding lower bounds were investigated in [34]. Other researchers have also investigated the use of existing communications infrastructure to enable these measurements, including LTE [35], 5G [36], and Wi-Fi [37]. Neural networks have also been considered for large networks of ToA-based localization sensors [38].

ToA estimation is well characterized by different types of bounds. Ziv–Zakai and Weiss–Weinstein lower bounds on ToA estimation were derived and discussed in [39] and [40], respectively, while a Cramér–Rao lower bound on hybrid time-of-arrival/received signal strength (ToA/RSS) was derived in [41]. Notably, 2D Cramér–Rao lower bounds on localization were presented in [42], and optimal sensor placement for time-difference-of-arrival (TDoA) systems is discussed in [43]. The CRLBs proposed in this work extend this

2D positioning bound to three dimensions, explicitly includes the number and placement of antennas in the bound, and additionally considers orientation estimation.

### 1.2. Contributions

In this manuscript, we make the following contributions:

- Derive a novel, closed-form, tractable CRLB on position estimation in TWR systems;
- Derive a novel, closed-form, tractable CRLB on orientation estimation in TWR systems;
- Implement the proposed CRLBs in a simple MATLAB simulation platform;
- Benchmark several popular estimators against the proposed CRLBs;
- Discuss how these results can directly inform aircraft design decisions.

## 2. Two-Way Ranging (TWR) Overview

In this section, we briefly define a timing and propagation model and outline some rudimentary time-of-arrival (ToA) and time-of-flight (ToF) estimation techniques.

### 2.1. Timing Model

Two-way ranging (as the name implies) involves a cooperative exchange between at least two users. Consider two users labeled $A$ and $B$ separated in space and operating with independent, imperfect clocks. At any given instant $n$, there is an offset between these clocks labeled $T^{(n)}$ and a distance between them $d^{(n)}$. If these users are interacting using over-the-air electromagnetic waveforms, then each transmission takes some time $\tau^{(n)} = d^{(n)}/c$ to propagate between the platforms, where $c$ is the speed of light. These interactions are depicted in Figure 2. The clock offsets $T$ and propagation delays $\tau$ are the fundamental quantities of interest and can be estimated by measuring the time-of-arrival (ToA) of each reception and applying an appropriate time-of-flight (ToF) estimation algorithm.

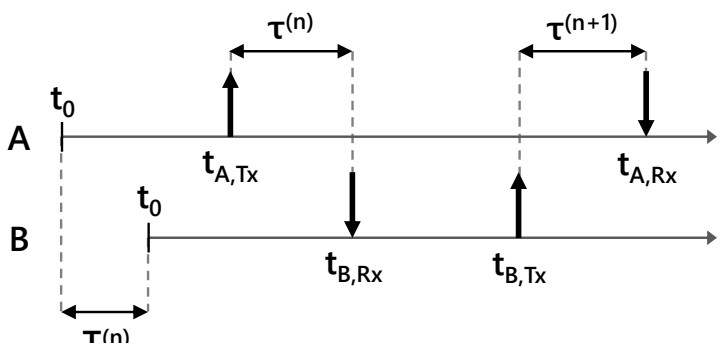

**Figure 2.** Depiction of two interactions between radios $A$ and $B$. Interactions are indexed by $n$. The time offset is labeled $T^{(\cdot)}$ and the propagation delay is labeled $\tau^{(\cdot)}$. Transmit events are labeled with up arrows, and corresponding receive events are labeled with down arrows. The clocks are misaligned, so the receive timestamps $t^{(\cdot)}_{(\cdot),Rx}$ follow Equations (1) and (2).

User $A$ transmits a waveform at time instant $n$, labeled $t^{(n)}_{A,Tx}$. This waveform takes a small amount of time to propagate to user $B$, labeled $\tau^{(n)}$. User $B$ receives the waveform at time $t^{(n)}_{B,Rx}$. If the clocks were perfectly aligned, the receive timestamp would simply be $t^{(n)}_{A,Tx} + \tau^{(n)}$. Because the clocks are misaligned, user $B$ measures the event earlier with respect to clock $B$, thus the received timestamp, as perceived by user $B$, is

$$t^{(n)}_{B,Rx} = t^{(n)}_{A,Tx} + \tau^{(n)} - T^{(n)}. \tag{1}$$

At some later time instant $n + 1$, user $B$ transmits a waveform to user $A$. The received timestamp, as perceived by user $A$ is

$$t_{A,Rx}^{(n+1)} = t_{B,Tx}^{(n+1)} + \tau^{(n+1)} + T^{(n+1)}. \tag{2}$$

### 2.2. Propagation Model

A signal $x$ is transmitted through a line-of-sight channel, during which it is distorted before the signal $z$ is received. The traditional propagation model [20] for this channel is

$$z = ax(t - \tau)e^{j2\pi f_c \tau}, \tag{3}$$

where $a$ is the complex channel attenuation, $\tau$ is the propagation delay, and $f_c$ is the carrier frequency. We assume a basic line-of-sight channel attenuation [44]

$$a^2 = \left( \frac{\lambda}{4\pi d} \right)^2 G_{Tx} G_{Rx}, \tag{4}$$

where $G_{Tx}$ and $G_{Rx}$ are the transmitter and receiver antenna gains, $\lambda$ is the signal wavelength, and $d$ is the distance between the two users.

Advanced ToA estimators require a much higher fidelity propagation model, which was previously explored in significant detail in [45]. The simplified model presented in Equation (3), however, is sufficient to motivate the following discussions and for the remainder of this manuscript.

### 2.3. Time-of-Arrival Estimation

Equations (1) and (2) form a system of two equations with two unknowns: $T$ and $\tau$. The transmit timestamps $t_{Tx}$ are known quantities, and the receive timestamps $t_{Rx}$ can be estimated using any number of ToA estimation techniques. ToA estimation is itself a rich field with both legacy and emerging solutions, so we briefly summarize the basic concept here and direct the reader to some relevant publications [40,45,46] for further reading.

Consider the correlation between received signal $z(t)$ and known transmitted signal $x(t)$:

$$g(\tau') = \left| \int dt \ z(t)x(t - \tau') \right|^2, \tag{5}$$

where $\tau'$ is a time delay relative to some time reference (analogous but not equivalent to the $\tau$ described above). By inspection, this correlation is maximized when the signals are aligned in time, so by defining an arbitrary but fixed reference time $t_0$, the local maximum likelihood (ML) ToA estimate is simply

$$\hat{t}_{Rx} = \hat{\tau}' - t_0 \ ; \ \hat{\tau}' = \arg\max_{\tau'} g(\tau'). \tag{6}$$

This formulation may seem pedantic for the limited scope of this discussion, but it is important for building the more advanced estimators outlined in [40,45,46]. There are numerous extensions to this simple estimator, including computationally efficient hardware implementations [5], leveraging phase information to adjust the estimate [45], and various iterative refinement methods [40,46].

### 2.4. Time-of-Flight Estimation

Equations (1) and (2) form a system of 2 equations with 2 unknowns $T$ and $\tau$. If the known transmit timestamps and estimated receive timestamps are shared between users, we have enough information to trivially solve the system. In reality, however, the clocks driving these devices can drift and the platforms may be in motion, so $T$ and $\tau$ may change between frames, thereby creating a system of two equations with four unknowns. For reasonable frame rates ($\geq$10 Hz) and modern RF hardware, both $T$ and $\tau$ are well

approximated by simple first- or second-order Markov models. By injecting these models and collecting data over multiple frames, we can construct closed-form estimators [47] or even Kalman tracking filters [5] to estimate $T$ and $\tau$. Various design implications may inform your choice of algorithm, but fundamentally, they all estimate $\tau$, from which we can infer the distance between platforms $d$. For multi-antenna systems, we can estimate the distance $d_{i,j}$ between each transmit antenna $i$ and receive antenna $j$, creating spatial diversity that enables relative position and orientation estimation. This configuration is depicted in Figure 3.

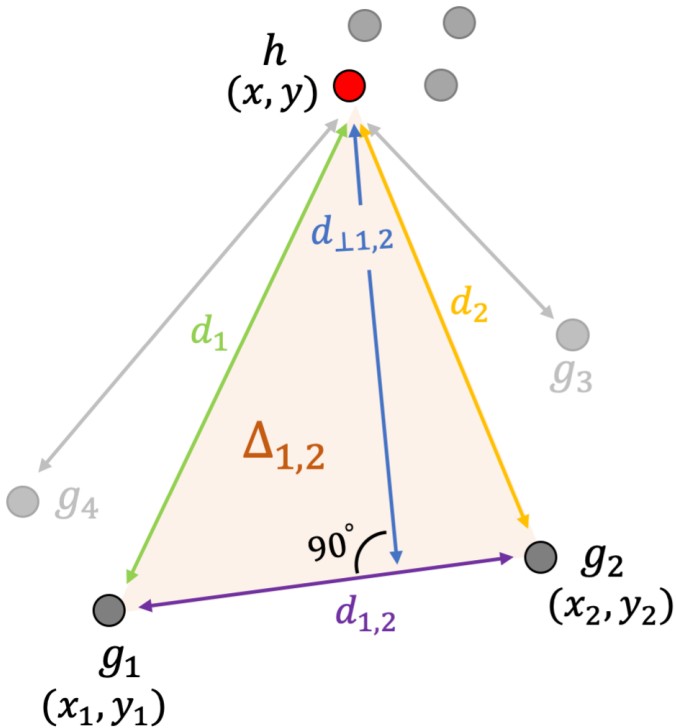

**Figure 3.** Example system configuration with a 4-antenna user $G$ and a 4-antenna user $H$. The distances between each antenna pair are extracted from time-of-flight (ToF) estimates produced by a two-way ranging (TWR) system.

## 3. Bounds on Position Estimation

In this section, we define the ranging model, derive 2D and 3D CRLBs on position estimation, and discuss geometric dilution of precision (GDoP).

### 3.1. Ranging Model

Consider $N$ ground antenna labeled $g_i$, which estimates the ToF $\tau_i$ between a target antenna $h$, depicted in Figure 3. Assume that the ToF estimates $\tau_i$ are unbiased estimators of the corresponding distance $d_i$, such that

$$\tau_i \sim \mathcal{N}(d_i/c, \sigma_{\tau_i}^2), \tag{7}$$

where $c$ is the speed of light, $\mathcal{N}$ denotes the normal distribution, $\sigma_{\tau_i}^2$ denotes the variance of the ToF estimator, and $d_i$ is the Euclidean distance between $g_i$ and $h$. The CRLBs for time-of-flight (ToF) estimation are well studied [45] and primarily depend on the integrated signal-to-noise ratio (ISNR) and the signal bandwidth. ISNR is a function of numerous variables, but if the bulk distance between platforms $G$ and $H$ is large compared with the local distance between antennas $g_i$, it is reasonable to assume that each link has comparable ISNR and, therefore, comparable variance, i.e., $\sigma_{\tau_i}^2 \approx \sigma_\tau^2 \ \forall i \ \{1, 2, \cdots, N\}$. This approximation is reasonable for most applications, but a notable counter-example is any

near-field application where the antennas of one system surround the antennas of the other, for example, the Airbus DeckFinder system [48].

The variance of an unbiased estimator $\hat{\boldsymbol{\alpha}} = [\hat{\alpha}_1 \, \hat{\alpha}_2 \, \cdots \, \hat{\alpha}_q]^T$ is bounded by $\sigma_{\hat{\alpha}_i} \geq [\mathbf{I}^{-1}(\alpha)]_{ii}$ where $\mathbf{I}(\alpha)$ is the Fisher information matrix (FIM) of size $q \times q$ and is defined [49] as

$$[\mathbf{I}(\alpha)]_{ij} = -E\left[\frac{\partial^2}{\partial\alpha_i\partial\alpha_j} \ln p(z;\alpha)\right]. \tag{8}$$

We use this definition of the CRLB for the remainder of this manuscript.

*3.2. Two-Dimensional CRLB on Position*

The probability density function of the ToF distribution summarized in Equation (7) is written explicitly as

$$p_\tau(\tau;h) = \frac{1}{\sqrt{2\pi\sigma_\tau^2}} \exp\left\{-\frac{1}{2\sigma_\tau^2}\sum_{i=1}^{N}\left(\tau - \frac{d_i}{c}\right)^2\right\}, \tag{9}$$

where $d_i = \sqrt{(x-x_i)^2 + (y-y_i)^2}$ is the Euclidean distance between antennae $g_i = [x_i, y_i]$ and $h = [x, y]$. The elements of the FIM $\mathbf{I}(h)$ are computed using Equation (8) as

$$[\mathbf{I}(h)]_{11} = \frac{1}{c^2\sigma_\tau^2}\sum_{i=1}^{N}\frac{(x-x_i)^2}{d_i^2},$$

$$[\mathbf{I}(h)]_{12} = [\mathbf{I}(\boldsymbol{\theta})]_{21} = \frac{1}{c^2\sigma_\tau^2}\sum_{i=1}^{N}\frac{(x-x_i)(y-y_i)}{d_i^2}, \tag{10}$$

$$[\mathbf{I}(h)]_{22} = \frac{1}{c^2\sigma_\tau^2}\sum_{i=1}^{N}\frac{(y-y_i)^2}{d_i^2}.$$

By definition, the CRLB on estimating the 2-D position of $h$ is $\sigma_h^2 \geq \mathbf{I}^{-1}(h) = \text{adj}\{\mathbf{I}(h)\}/|\mathbf{I}(h)|$, where the adjugate adj$\{.\}$ and determinant $|.|$ are written as

$$\text{adj}\{\mathbf{I}(h)\} = \frac{1}{c^2\sigma_\tau^2}\begin{bmatrix} \sum_{i=1}^{N}\frac{(y-y_i)^2}{d_i^2} & -\sum_{i=1}^{N}\frac{(x-x_i)(y-y_i)}{d_i^2} \\ -\sum_{i=1}^{N}\frac{(x-x_i)(y-y_i)}{d_i^2} & \sum_{i=1}^{N}\frac{(x-x_i)^2}{d_i^2} \end{bmatrix}, \tag{11}$$

$$|\mathbf{I}(h)| = \frac{2}{(c^2\sigma_\tau^2)^2}\sum_{\substack{i=1 \\ i\neq j}}^{N}\sum_{j=1}^{N}\left(\frac{\mathcal{D}_2}{2d_id_j}\right)^2, \tag{12}$$

where $\mathcal{D}_2$ is the determinant of the distance matrix

$$\mathcal{D}_2 = \begin{vmatrix} x-x_i & y-y_i \\ x-x_j & y-y_j \end{vmatrix}. \tag{13}$$

To make this result more accessible to a design engineer, we can loosely visualize the CRLB as a scalar value by defining an overall "position variance" as $\sigma_p^2 = \sigma_x^2 + \sigma_y^2 \geq \text{tr}\{\mathbf{I}^{-1}(h)\}$, which can be written as

$$\sigma_p^2 \geq \frac{c^2\sigma_\tau^2 N}{2}\left[\sum_{\substack{i=1 \\ i\neq j}}^{N}\sum_{j=1}^{N}\left(\frac{\mathcal{D}_2}{2\,d_i\,d_j}\right)^2\right]^{-1}. \tag{14}$$

### 3.3. Three-Dimensional CRLB on Position

Reconsider Equation (9) with the 3D definitions $d_i = \sqrt{(x - x_i)^2 + (y - y_i)^2 + (z - z_i)^2}$, $g_i = [x_i, y_i, z_i]$, and $h = [x, y, z]$. The FIM now takes the form

$$\mathbf{I}(h) = \frac{1}{c^2 \sigma_\tau^2} \begin{bmatrix} \sum_{i=1}^{N} \frac{(x-x_i)^2}{d_i^2} & \sum_{i=1}^{N} \frac{(x-x_i)(y-y_i)}{d_i^2} & \sum_{i=1}^{N} \frac{(x-x_i)(z-z_i)}{d_i^2} \\ \sum_{i=1}^{N} \frac{(x-x_i)(y-y_i)}{d_i^2} & \sum_{i=1}^{N} \frac{(y-y_i)^2}{d_i^2} & \sum_{i=1}^{N} \frac{(y-y_i)(z-z_i)}{d_i^2} \\ \sum_{i=1}^{N} \frac{(x-x_i)(z-z_i)}{d_i^2} & \sum_{i=1}^{N} \frac{(y-y_i)(z-z_i)}{d_i^2} & \sum_{i=1}^{N} \frac{(z-z_i)^2}{d_i^2} \end{bmatrix}. \tag{15}$$

The CRLB is again expressed as $\sigma_{\hat{h}}^2 \geq \mathbf{I}^{-1}(h) = \text{adj}\{\mathbf{I}(h)\}/|\mathbf{I}(h)|$, where the adjugate now takes the form $\text{adj}\{\mathbf{I}(h)\} = \frac{2}{(c^2 \sigma_\tau^2)^2} \mathbf{A}$, where

$$\mathbf{A} = \begin{bmatrix} \sum_{i=1}^{N} \sum_{j=1}^{N} \frac{V_{i,j}(y,z)^2}{(d_i d_j)^2} & \sum_{i=1}^{N} \sum_{j=1}^{N} \frac{V_{i,j}(x,z) V_{i,j}(z,y)}{(d_i d_j)^2} & \sum_{i=1}^{N} \sum_{j=1}^{N} \frac{V_{i,j}(x,y) V_{i,j}(y,z)}{(d_i d_j)^2} \\ \sum_{i=1}^{N} \sum_{j=1}^{N} \frac{V_{i,j}(x,z) V_{i,j}(z,y)}{(d_i d_j)^2} & \sum_{i=1}^{N} \sum_{j=1}^{N} \frac{V_{i,j}(x,z)^2}{(d_i d_j)^2} & \sum_{i=1}^{N} \sum_{j=1}^{N} \frac{V_{i,j}(y,x) V_{i,j}(x,z)}{(d_i d_j)^2} \\ \sum_{i=1}^{N} \sum_{j=1}^{N} \frac{V_{i,j}(x,y) V_{i,j}(y,z)}{(d_i d_j)^2} & \sum_{i=1}^{N} \sum_{j=1}^{N} \frac{V_{i,j}(y,x) V_{i,j}(x,z)}{(d_i d_j)^2} & \sum_{i=1}^{N} \sum_{j=1}^{N} \frac{V_{i,j}(x,y)^2}{(d_i d_j)^2} \end{bmatrix}, \tag{16}$$

and the determinant takes the form

$$|\mathbf{I}(h)| = \frac{6}{(c^2 \sigma_\tau^2)^3} \sum_{i=1}^{N} \sum_{j=1}^{N} \sum_{k=1}^{N} \left( \frac{V_{i,j,k}}{d_i d_j d_k} \right)^2, \tag{17}$$

where $V_{i,j,k}$ is the volume of a tetrahedron formed by points $h$, $g_i$, $g_j$, and $g_k$, as depicted in Figure 4. When represented as the Cayley–Menger determinant, this takes the form

$$V_{i,j,k} = \frac{1}{6} \begin{vmatrix} x - x_i & y - y_i & z - z_i \\ x - x_j & y - y_j & z - z_j \\ x - x_k & y - y_k & z - z_k \end{vmatrix}, \tag{18}$$

Thus, the closed-form CRLB is written as

$$\mathbf{I}(h)^{-1} = \frac{\text{adj}\{\mathbf{I}(h)\}}{|\mathbf{I}(h)|} = \frac{c^2 \sigma_\tau^2}{3} \mathbf{A} \left[ \sum_{i=1}^{N} \sum_{j=1}^{N} \sum_{k=1}^{N} \left( \frac{V_{i,j,k}}{d_i d_j d_k} \right)^2 \right]^{-1} \tag{19}$$

As in Equation (14), we write the "position variance" $\sigma_p^2 = \sigma_x^2 + \sigma_y^2 + \sigma_z^2 \geq \text{tr}\{\mathbf{I}^{-1}(h)\}$ as

$$\sigma_p^2 \geq \frac{c^2 \sigma_\tau^2}{3} \frac{\left[ \sum_{i=1}^{N} \sum_{j=1}^{N} \frac{V_{i,j}(x,y)^2 + V_{i,j}(y,z)^2 + V_{i,j}(x,z)^2}{(d_i d_j)^2} \right]}{\left[ \sum_{i=1}^{N} \sum_{j=1}^{N} \sum_{k=1}^{N} \left( \frac{V_{i,j,k}}{d_i d_j d_k} \right)^2 \right]} \tag{20}$$

where $V_{i,j}(\cdot, \cdot)$ is the area of the triangle in three-dimensional Cartesian space with vertices $h$, $g_i$, and $g_j$ when projected onto the $(\cdot, \cdot)$ plane, i.e.,

$$V_{i,j}(x,y) = \frac{1}{2} \begin{vmatrix} x - x_i & y - y_i \\ x - x_j & y - y_j \end{vmatrix}, \ V_{i,j}(y,z) = \frac{1}{2} \begin{vmatrix} y - y_i & z - z_i \\ y - y_j & z - z_j \end{vmatrix}, \ V_{i,j}(x,z) = \frac{1}{2} \begin{vmatrix} x - x_i & z - z_i \\ x - x_j & z - z_j \end{vmatrix}$$

To the best of our knowledge, this is the first closed-form, three-dimensional CRLB on position estimation in a TWR system.

*3.4. Geometric Interpretation of D*

The determinant $\mathcal{D}_2$ can be interpreted in several different ways:

1. The shortest (or perpendicular) distance between the target antenna $h = [x, y]$ and the line joining ground node antennae $g_i = [x_i, y_i]$ and $g_j = [x_j, y_j]$ is $d_{\perp i,j} = \mathcal{D}_2/d_{i,j}$, where $d_{i,j} = d(g_i, g_j)$ (refer to Figure 3). The lower bound then reduces [50] to

$$\sigma_p^2 \geq \frac{c^2 \sigma_\tau^2 N}{2} \left[ \sum_{\substack{i=1 \\ i \neq j}}^{N} \sum_{j=1}^{N} \left( \frac{d_{\perp i,j}\, d_{i,j}}{2\, d_i\, d_j} \right)^2 \right]^{-1}. \tag{21}$$

2. As a higher-order generalization, interpret $\mathcal{D}_2/2!$ as the volume of a 2-simplex, $V_{i,j}$, formed by the antennae $h$, $g_i$, and $g_j$ (refer to Figure 3). The bound then becomes

$$\sigma_p^2 \geq \frac{c^2 \sigma_\tau^2 N}{2!} \left[ \sum_{\substack{i=1 \\ i \neq j}}^{N} \sum_{j=1}^{N} \left( \frac{V_{i,j}}{d_i\, d_j} \right)^2 \right]^{-1}. \tag{22}$$

3. The quantity $(d_{\perp i,j}\, d_{i,j}/d_i\, d_j)$ is called geometric conditioning $\mathcal{A}_{i,j}$ and is a measure of the area of a parallelogram contained by vectors $\overrightarrow{hg_i}$ and $\overrightarrow{hg_j}$, scaled by length of those vectors. In this case, the lower bound becomes

$$\sigma_p^2 \geq \frac{c^2 \sigma_\tau^2 N}{2} \left[ \sum_{\substack{i=1 \\ i \neq j}}^{N} \sum_{j=1}^{N} \left( \frac{\mathcal{A}_{i,j}}{2} \right)^2 \right]^{-1}, \tag{23}$$

where $\mathcal{A}_{i,j}/2$ can be envisioned as the area of triangle $\Delta_{i,j}$ enclosed by unit vectors in the same direction. It is interesting to note that this geometric conditioning is independent of the absolute distances between the ground and target antennae.

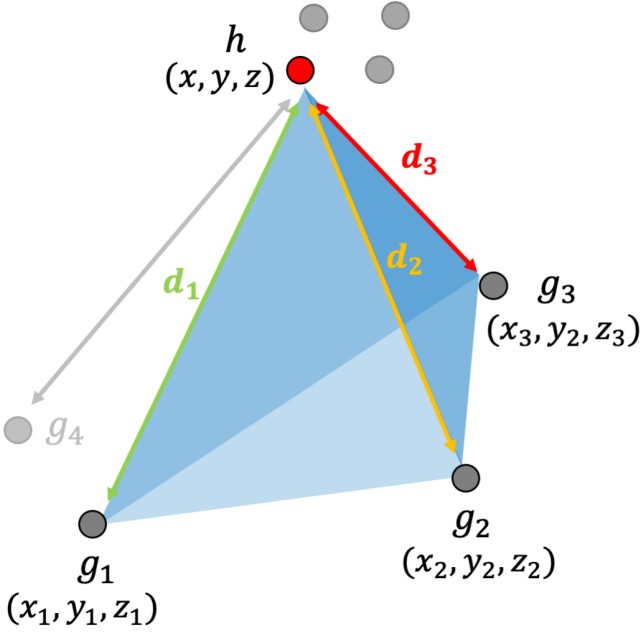

**Figure 4.** Geometric interpretation of the volume of the tetrahedron $V_{i,j,k}(d_i, d_j, d_k)$, indicated in blue.

### 3.5. Geometric Dilution of Precision

By inspection, the CRLB derived in Equation (20) only depends on the ToF estimator performance ($\sigma_\tau$) and the geometry of the ground nodes $g_i$. In related estimation problems, the geometric portion of this equation is often referred to as "geometric dilution of precision", and effectively represents the penalty incurred by using a noisy estimate to further estimate other quantities, in this case using range to estimate position. We can rewrite Equation (20) in the form $\sigma_p^2 \geq c^2 \sigma_\tau^2 \, \text{GDoP}$, where geometric dilution of precision (GDoP) is a unit-less quantity defined by

$$\text{GDoP} = \sqrt{\frac{N}{2 \sum_{\substack{i=1 \\ i \neq j}}^{N} \sum_{j=1}^{N} \Delta_{i,j}^2}} = \sqrt{\frac{2\,N}{\sum_{k=1}^{M} \sin^2(\gamma_k)}}, \tag{24}$$

where $\Delta_{i,j} = \frac{1}{2} \sin(\gamma_{i,j})$ is the area of a triangle with unit sides and angle $\angle \gamma_{i,j}$ between them and $M = \binom{N}{2}$ is the number of unique triangles enclosed by $\overrightarrow{hg_i}$ and $\overrightarrow{hg_j}$ $\forall\, i, j \in \{1, 2, \cdots, N\}$. By inspection, observe that maximizing $\mathcal{M} = \sum_{k=1}^{M} \sin^2(\gamma_k)$ minimizes the CRLB for a given ranging precision $\sigma_\tau$. This is a convex problem within the range $\gamma_k \in (0, \pi]$ with unique unambiguous solutions $\{\gamma_1, \gamma_2, \cdots, \gamma_M\}$ that yield the best performance for a given number of ground antennae $N$. This is important because it directly informs the optimal antenna placement for a fixed number of antennas $N$.

In Figure 5, the positioning CRLB, from Equation (20) is plotted as a function of the ranging precision $\sigma_\tau$ and the geometric dilution of precision from Equation (24). By reformulating the bound as a function of traditional metrics (ranging precision and GDoP) rather than abstract volumes, this formulation provides a visual reference of the trade space between the quantities that a design engineer actually cares about, namely, the ranging precision, the positioning precision, and the GDoP induced by the platform geometry.

The ranging precision $\sigma_\tau$ is primarily governed by the choice of ranging technology. If the design constraints limit the available locations of the antennas on an aircraft, then this curve can help inform decisions about which ranging technology is most appropriate to achieve a target positioning precision. In contrast, if the ranging precision is fixed either by design or operating conditions, this curve can instead inform decisions about the optimal placement of antennas on an aircraft or ground installation. Additional insight in this regard is also provided in [43]. Naturally, all of this information is contained in the original formulation presented in Equation (20), this figure is simply included as a visual reference that does not require any calculations.

In Table 1, the best achievable GDoP value is enumerated as a function of the number of antennas $N$ between three and eight. This particular positioning method requires at least three antennas to converge on a solution and experiences diminishing returns beyond eight. In Table 2, we include a qualitative description of GDoP values summarized from [51]. In many TWR and other positioning applications—particularly GPS—it is common to summarize the GDoP of an installation without expressly stating the actual geometry, so this information is included as a coarse reference.

**Table 1.** Best achievable GDoP vs. number of antennas.

| N | 3 | 4 | 5 | 6 | 7 | 8 |
|---|---|---|---|---|---|---|
| Best GDoP | 1.155 | 1.0 | 0.894 | 0.816 | 0.756 | 0.707 |

**Table 2.** Qualitative rating of realistic GDoP values.

| GDoP | 1 | 2–3 | 4–6 | 7–8 | 9–20 | 21–50 |
|------|---|-----|-----|-----|------|-------|
| "Rating" | "Ideal" | Excellent | Good | Moderate | Fair | Poor |

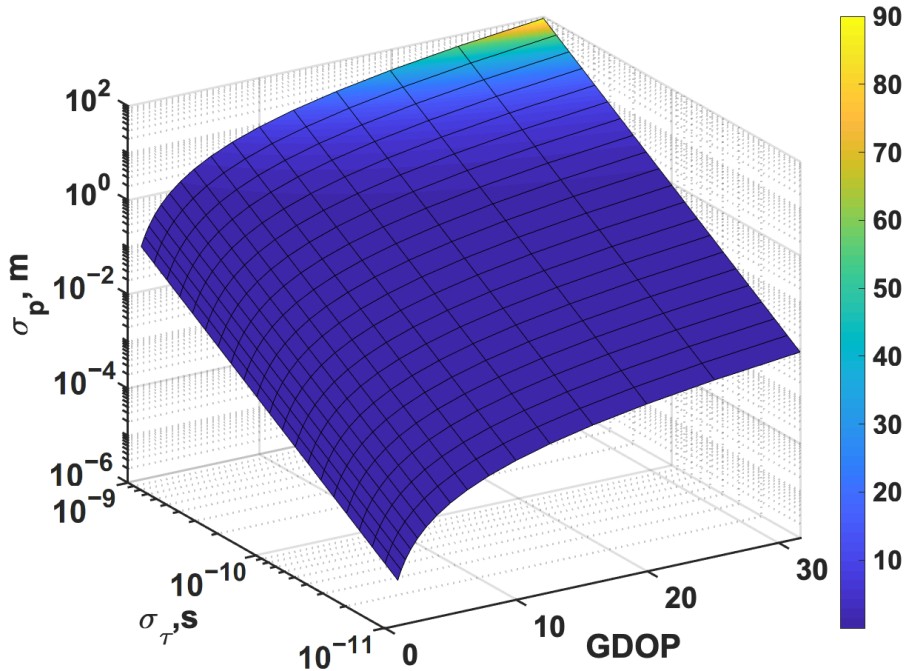

**Figure 5.** Performance manifold defined by Equation (20) as a function of ranging precision and GDoP for N = 4.

## 4. Bounds on Orientation Estimation

In this section, we derive 2-dimensional (2D) and 3-dimensional (3D) CRLBs for an orientation estimator based on the position estimation described process in the previous section.

### 4.1. Model

In the previous section, we derived bounds on the estimation of the position of antennas $h_i$ with respect to reference antennas $g_i$. For N target antennas, we form $M = \binom{N}{2}$ vectors labeled $\overline{h_i h_j} \; \forall \, i, j \in \{1, 2, \cdots, N\}$ and $i \neq j$. We now use these estimated positions to estimate the relative orientation of platform $H$ with respect to platform $G$ by observing the rotation of the vectors $\overline{h_i h_j}$ relative to $\overline{g_i g_j}$. Consider the normalized vectors

$$\overline{b}_k = \overline{h_i h_j} / \left\| \overline{h_i h_j} \right\|^{1/2}, \; \overline{v}_k = \overline{g_i g_j} / \left\| \overline{g_i g_j} \right\|^{1/2}, \; k \in \{1, 2, \cdots, M\} \tag{25}$$

We assume [52] that the unit vectors $\overline{b}_k$ follow a von Mises distribution with mean direction $\overline{\mu}_k$ and concentration $\kappa_k$, i.e.,

$$\overline{b}_k \sim \mathcal{M}(\overline{\mu}_k, \kappa_k) \quad ; \quad \overline{\mu}_k = \mathbf{R} \overline{v}_k, \tag{26}$$

where $\kappa \geq 0$, $\|\overline{\mu}\| = 1$, and $\mathbf{R}$ is a proper rotation matrix. For $\kappa \ll 1$ the von Mises distribution reduces to a uniform distribution in which $\overline{b}_k$ are uniformly distributed on a circle irrespective of the rotation matrix $\mathbf{R}$. For all other scenarios, if the variance in estimating position $h_i$ and $h_j$ is $\sigma_{h_i}$ and $\sigma_{h_j}$, we consider the circular variance in $\overline{b}_k$ as

$\sigma_{\bar{b}_k}^2 = \sigma_{h_i}^2 + \sigma_{h_j}^2$; hence, the mean resultant length $\rho_k$ of the wrapped distribution and the concentration factor $\kappa_k$ are given by

$$\rho_k = e^{-\sigma_{h_i h_j}^2/2} \quad ; \quad \kappa_k = A^{-1}(\rho_k) \tag{27}$$

where $A(\cdot)$ is a ratio of modified Bessel functions.

### 4.2. Two-Dimensional CRLB on Orientation

The probability density function [52] of 2D vectors $\bar{b} = [\bar{b}_1, \bar{b}_2, \cdots, \bar{b}_M]$ with mean $\mu_k = \mathbf{R}(\theta)\bar{v}_k$, relative orientation $\theta$, proper 2D rotation matrix $\mathbf{R}(\theta)$, ground node vectors $\bar{v} = [\bar{v}_1, \bar{v}_2, \cdots, \bar{v}_M]$, and concentration $\kappa = [\kappa_1, \kappa_2, \cdots, \kappa_M]$ is

$$p_{\bar{b}}(\bar{b}; \bar{v}, \theta, \kappa) = \left( \prod_{k=1}^{M} \frac{1}{2\pi I_0(\kappa_k)} \right) \exp\left\{ \sum_{k=1}^{M} \kappa_k \, \bar{b}_k^T \mathbf{R}(\theta) \bar{v}_k \right\} \tag{28}$$

where the normalizing constant $I_0(\cdot)$ is a modified Bessel function of the first kind at order 0. The FIM $\mathbf{I}(\theta)$ is computed as

$$\mathbf{I}(\theta) = -\mathrm{E}\left[ \frac{\partial^2}{\partial \theta^2} \ln f(\bar{v}; \bar{v}, \theta, \kappa) \right] = \sum_{k=1}^{M} \kappa_k \, \bar{b}_k^T \mathbf{R}(\theta) \bar{v}_k, \tag{29}$$

since $\mathbf{R}''(\theta) = -\mathbf{R}(\theta)$. Thus, the CRLB on an unbiased orientation estimator $\hat{\theta}$ is

$$\sigma_\theta^2 \geq \mathbf{I}^{-1}(\theta) = \frac{1}{\sum_{k=1}^{M} \kappa_k \, \bar{b}_k^T \mathbf{R}(\theta) \bar{v}_k} \tag{30}$$

It is important to note that the lower bound is independent of the rotation angle $\theta$ and only depends on the concentration factor $\kappa$ and the number of observations $M$.

### 4.3. Three-Dimensional CRLB on Orientation

Assume the 3D observations $\bar{b}_k$ follow a von Mises distribution [52] whose probability density function is given by

$$p_{\bar{b}}(\bar{b}; \bar{v}, \theta, \kappa) = \left( \prod_{k=1}^{M} C_3(\kappa_k) \right) \exp\left\{ \sum_{k=1}^{M} \kappa_k \, \bar{b}_k^T \mathbf{R}(\theta) \bar{v}_k \right\} \tag{31}$$

where concentrations $\kappa_k \geq 0$, means $\|\bar{\mu}_k\| = 1$, $\bar{\mu}_k = \mathbf{R}(\theta)\bar{v}_k$, and $C_3(\kappa)$ is a normalized constant

$$C_3(\kappa) = \frac{\kappa}{4\pi \sinh \kappa} = \frac{\kappa}{4\pi(e^\kappa - e^{-\kappa})} \tag{32}$$

The relative rotation between two 2D vectors can be sufficiently defined with a single angle, whereas at least three are required to do the same in a 3D space. For convenience, we chose Tait Bryan angles—roll ($\alpha$), pitch ($\beta$) and yaw ($\gamma$)—to represent the relative attitude of the target $H$ with respect to the observing node $G$. Therefore, the rotation matrix $\mathbf{R}(\theta)$ can be written as the product of three independent rotations by $\alpha, \beta, \gamma$ about $x, y, z$ axis, i.e., $\mathbf{R}(\theta) = \mathbf{R}_z(\gamma) \mathbf{R}_y(\beta) \mathbf{R}_x(\alpha)$ where $\theta = [\alpha, \beta, \gamma]$, which becomes

$$\mathbf{R}(\theta) = \begin{bmatrix} \cos(\beta)\cos(\gamma) & \begin{matrix} -\cos(\alpha)\sin(\gamma) + \\ \sin(\alpha)\sin(\beta)\cos(\gamma) \end{matrix} & \begin{matrix} \sin(\alpha)\sin(\gamma) + \\ \cos(\alpha)\sin(\beta)\cos(\gamma) \end{matrix} \\ \cos(\beta)\sin(\gamma) & \begin{matrix} \cos(\alpha)\cos(\gamma) + \\ \sin(\alpha)\sin(\beta)\sin(\gamma) \end{matrix} & \begin{matrix} -\sin(\alpha)\cos(\gamma) + \\ \cos(\alpha)\sin(\beta)\sin(\gamma) \end{matrix} \\ -\sin(\beta) & \sin(\alpha)\cos(\beta) & \cos(\alpha)\cos(\beta) \end{bmatrix}. \tag{33}$$

The FIM $\mathbf{I}(\boldsymbol{\theta}) = -\mathrm{E}\left[\dfrac{\partial^2}{\partial \boldsymbol{\theta}^2} \ln p_{\overline{b}}(\overline{\boldsymbol{b}}; \overline{\boldsymbol{v}}, \boldsymbol{\theta}, \boldsymbol{\kappa})\right]$ reduces to

$$
\mathbf{I}(\boldsymbol{\theta}) = \begin{bmatrix}
-\sum_{i=1}^{N} \kappa_i\, b_i^{\,T} \frac{\partial^2}{\partial \alpha^2} \mathbf{R}(\boldsymbol{\theta}) \bar{v}_i & -\sum_{i=1}^{N} \kappa_i\, b_i^{\,T} \frac{\partial^2}{\partial \alpha \partial \beta} \mathbf{R}(\boldsymbol{\theta}) \bar{v}_i & -\sum_{i=1}^{N} \kappa_i\, b_i^{\,T} \frac{\partial^2}{\partial \alpha \partial \gamma} \mathbf{R}(\boldsymbol{\theta}) \bar{v}_i \\
-\sum_{i=1}^{N} \kappa_i\, b_i^{\,T} \frac{\partial^2}{\partial \alpha \partial \beta} \mathbf{R}(\boldsymbol{\theta}) \bar{v}_i & -\sum_{i=1}^{N} \kappa_i\, b_i^{\,T} \frac{\partial^2}{\partial \beta^2} \mathbf{R}(\boldsymbol{\theta}) \bar{v}_i & -\sum_{i=1}^{N} \kappa_i\, b_i^{\,T} \frac{\partial^2}{\partial \beta \partial \gamma} \mathbf{R}(\boldsymbol{\theta}) \bar{v}_i \\
-\sum_{i=1}^{N} \kappa_i\, b_i^{\,T} \frac{\partial^2}{\partial \alpha \partial \gamma} \mathbf{R}(\boldsymbol{\theta}) \bar{v}_i & -\sum_{i=1}^{N} \kappa_i\, b_i^{\,T} \frac{\partial^2}{\partial \beta \partial \gamma} \mathbf{R}(\boldsymbol{\theta}) \bar{v}_i & -\sum_{i=1}^{N} \kappa_i\, b_i^{\,T} \frac{\partial^2}{\partial \gamma^2} \mathbf{R}(\boldsymbol{\theta}) \bar{v}_i
\end{bmatrix}
$$

The CRLB is then computed by inverting the FIM,

$$
\mathbf{I}(\boldsymbol{\theta})^{-1} = \frac{\mathrm{adj}\{\mathbf{I}(h)\}}{|\mathbf{I}(h)|}. \tag{34}
$$

It is important to note that the structure of the rotation matrix provides simplifications that can be leveraged to derive the CRLB in closed form. The partial derivatives in the rotation matrix significantly complicate the computation of Equation (34), which can be mitigated by the following simplifications. Let $S[a, b, c]$ represent a skew-symmetric matrix of the form

$$
S[a, b, c] = \begin{bmatrix} 0 & -c & b \\ c & 0 & -a \\ -b & a & 0 \end{bmatrix}. \tag{35}
$$

Useful properties of $S$ include $\mathbf{S}[1, 0, 0]^2 = \mathrm{diag}[0, -1, -1]$, $\mathbf{S}[0, 1, 0]^2 = \mathrm{diag}[-1, 0, -1]$, and $\mathbf{S}[0, 0, 1]^2 = \mathrm{diag}[-1, -1, 0]$. The first-order partial derivatives in the rotation can be simplified using

$$
\frac{\partial}{\partial \alpha} \mathbf{R}_x(\alpha) = \mathbf{S}[1, 0, 0]\, \mathbf{R}_x(\alpha), \tag{36}
$$

$$
\frac{\partial}{\partial \beta} \mathbf{R}_y(\beta) = \mathbf{S}[0, 1, 0]\, \mathbf{R}_y(\beta), \tag{37}
$$

$$
\frac{\partial}{\partial \gamma} \mathbf{R}_z(\gamma) = \mathbf{S}[0, 0, 1]\, \mathbf{R}_z(\gamma), \tag{38}
$$

Therefore,

$$
\frac{\partial}{\partial \alpha} \mathbf{R}(\boldsymbol{\theta}) = \mathbf{R}_z(\gamma)\, \mathbf{R}_y(\beta)\, \mathbf{S}[1, 0, 0] \mathbf{R}_x(\alpha), \tag{39}
$$

$$
\frac{\partial}{\partial \beta} \mathbf{R}(\boldsymbol{\theta}) = \mathbf{R}_z(\gamma)\, \mathbf{S}[0, 1, 0] \mathbf{R}_y(\beta)\, \mathbf{R}_x(\alpha), \tag{40}
$$

$$
\frac{\partial}{\partial \gamma} \mathbf{R}(\boldsymbol{\theta}) = \mathbf{S}[0, 0, 1] \mathbf{R}_z(\gamma)\, \mathbf{R}_y(\beta)\, \mathbf{R}_x(\alpha), \tag{41}
$$

and the second-order derivatives of $\mathbf{R}(\boldsymbol{\theta})$ can be simplified to

$$
\frac{\partial^2}{\partial \alpha^2} \mathbf{R}(\boldsymbol{\theta}) = \mathbf{R}_z(\gamma)\, \mathbf{R}_y(\beta)\, \mathbf{S}[1, 0, 0]^2 \mathbf{R}_x(\alpha), \tag{42}
$$

$$
\frac{\partial^2}{\partial \beta^2} \mathbf{R}(\boldsymbol{\theta}) = \mathbf{R}_z(\gamma)\, \mathbf{S}[0, 1, 0]^2 \mathbf{R}_y(\beta)\, \mathbf{R}_x(\alpha), \tag{43}
$$

$$
\frac{\partial^2}{\partial \gamma^2} \mathbf{R}(\boldsymbol{\theta}) = \mathbf{S}[0, 0, 1]^2 \mathbf{R}_z(\gamma)\, \mathbf{R}_y(\beta)\, \mathbf{R}_x(\alpha), \tag{44}
$$

$$
\frac{\partial^2}{\partial \alpha \partial \beta} \mathbf{R}(\boldsymbol{\theta}) = \mathbf{R}_z(\gamma)\, \mathbf{S}[0, 1, 0] \mathbf{R}_y(\beta)\, \mathbf{S}[1, 0, 0] \mathbf{R}_x(\alpha), \tag{45}
$$

$$\frac{\partial^2}{\partial \beta \partial \gamma} \mathbf{R}(\boldsymbol{\theta}) = \mathbf{S}[0,0,1] \mathbf{R}_z(\gamma) \, \mathbf{S}[0,1,0] \mathbf{R}_y(\beta) \, \mathbf{R}_x(\alpha), \tag{46}$$

$$\frac{\partial^2}{\partial \alpha \partial \gamma} \mathbf{R}(\boldsymbol{\theta}) = \mathbf{S}[0,0,1] \mathbf{R}_z(\gamma) \, \mathbf{R}_y(\beta) \, \mathbf{S}[1,0,0] \mathbf{R}_x(\alpha). \tag{47}$$

To the best of our knowledge, this is also the first closed-form, three-dimensional CRLB on orientation estimation in a TWR system.

## 5. Simulation Results

In this section, we benchmark several common position and orientation estimators against the bounds derived in the previous sections on a MATLAB simulation platform. We also include a very brief discussion of position and orientation estimation for reference.

### 5.1. Estimation Preliminaries

For the given model, the expected log-likelihood function $\mathcal{L}(\theta) = \mathrm{E}[\ln p_\tau(t; h)]$ reduces to a least-square problem. We adopt the ordinary least squares (LS) formulation from [53] and present it here for the sake of completeness. Define a matrix $\mathbf{X}$ and elements of a column vector $\mathbf{y}$ for $i \in \{1, 2, \cdots, N\}$ as

$$\mathbf{X} = \begin{bmatrix} 2(x_1 - \bar{x}) & 2(y_1 - \bar{y}) \\ 2(x_2 - \bar{x}) & 2(y_2 - \bar{y}) \\ \vdots & \vdots \\ 2(x_N - \bar{x}) & 2(y_N - \bar{y}) \end{bmatrix} \quad ; \quad \mathbf{y}[i] = \bar{d}_i^2 - d_i^2 + \mathbf{X}\bar{g} \tag{48}$$

where $g_i = [x_i, y_i]^T$ is the location of $N$ ground node antennae, $\bar{g} = [\bar{x}, \bar{y}]^T$ is their mean location, and $\bar{d}_i = \|g_i - \bar{g}\|^{1/2}$ is the distance between $g_i$ and $\bar{g}$. The unbiased, linearized, least-square estimate of position $h$ is simply

$$\hat{h}_{\mathrm{LS}} = (\mathbf{X}^T \mathbf{X})^{-1} \mathbf{X}^T Y \tag{49}$$

For the given model, we propose a maximum-likelihood orientation estimator. The expected log-likelihood function $\mathcal{L}(\theta) = \mathrm{E}\left[\ln p_{\overline{\boldsymbol{b}}}(\overline{\boldsymbol{b}}; \overline{v}, \theta, \kappa)\right]$ is

$$\mathcal{L}(\theta) = \sum_{k=1}^{M} \kappa_k \, \overline{b}_k^T \, \mathbf{R}(\theta) \, \overline{v}_k + \text{constants}. \tag{50}$$

Since this is a convex function, there exists a unique, unambiguous angle $\theta \in [0, \pi]$ for which $\mathcal{L}(\theta)$ is maximized, which is computed as

$$\frac{d}{d\theta} \mathcal{L}(\theta) = \sum_{k=1}^{N} \kappa_k \, \overline{b}_k^T \mathbf{S}(\theta) \overline{v}_k = 0, \tag{51}$$

where $\mathbf{S}(\theta) = \frac{d}{d\theta} \mathbf{R}(\theta)$. Therefore, the maximum-likelihood estimate of orientation is obtained by solving Equation (51) as

$$\hat{\theta}_{ML} = \arctan\left\{ \frac{\sum_{k=1}^{M} \kappa_k \, \overline{b}_k^T \mathbf{R}(\frac{\pi}{2}) \, \overline{v}_k}{\sum_{k=1}^{M} \kappa_k \, \overline{b}_k^T \overline{v}_k} \right\} \tag{52}$$

It is well established that $\kappa = A^{-1}(\rho)$ and, for a two-dimensional space, $A(\cdot) = I_1(\cdot)/I_0(\cdot)$ is a ratio of modified Bessel functions of the first kind and order one and zero; there are multiple approximations to $A^{-1}(\cdot)$ [54], and we adopt the one provided in [55].

We attain a comprehensive measure for concentration factor $\kappa$ by assuming $\kappa_k = \kappa$ for $k \in \{1, 2, \cdots, M\}$, $\sigma_{h_i} = \sigma_p$ for $i \in \{1, 2, \cdots, N\}$, and approximating $\kappa$ as

$$\hat{\kappa} = \frac{\bar{\rho}\,(2 - \bar{\rho}^2)}{1 - \bar{\rho}^2} \quad ; \quad \bar{\rho} = e^{-\sigma_p^2}, \tag{53}$$

where $0 \leq \bar{\rho} \leq 1$ is the mean resultant length. When the standard deviation in position estimation $\sigma_p > 1$ meter, the concentration factor $\kappa \ll 1$. In those scenarios, the von Mises model assumption fails, and the vectors $\bar{b}_k$ do not contain any information regarding rotations $\theta$.

### 5.2. Position—3D CRLB

We implemented three position estimators in our MATLAB simulation platform: ordinary least squares (OLS); iteratively reweighted least squares (IRLS); and non-linear least squares (NLLS). We distributed $N = 50$ ground antennas randomly in a 10-meter sphere centered at the origin and placed the target antenna $h$ at (0, 0, 50) meters. We swept the distance estimator standard deviation $\sigma_d$ between $10^{-6}$ and 10 m, running 1000 Monte Carlo trials at each test point. These results are depicted in Figure 6. Each of the three estimators closely approaches the CRLB in the region of interest, with the NLLS estimator achieving slightly better performance than the IRLS and OLS estimators. This result demonstrates that the proposed CRLB is tractable for a massive number of sensing elements (50) and that it agrees with existing well-known estimators.

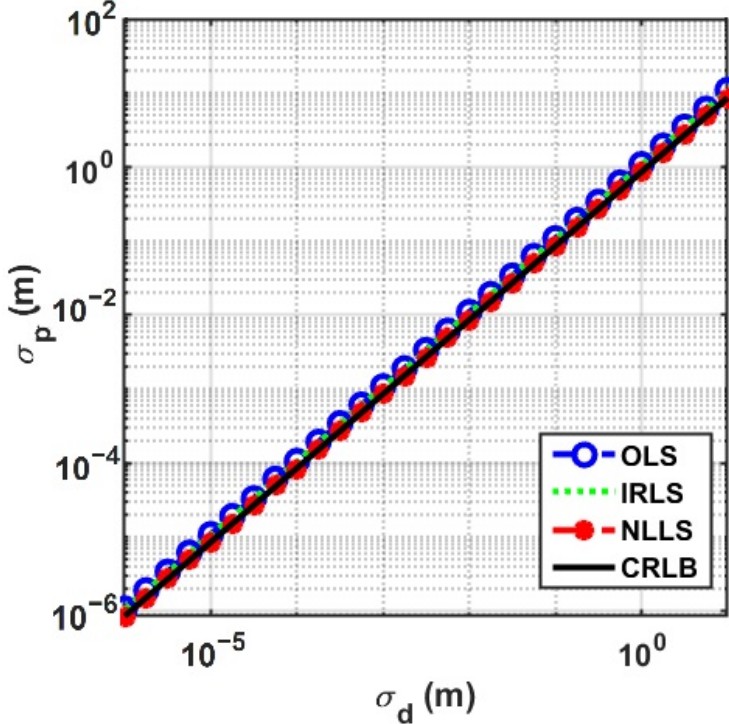

**Figure 6.** Performance comparison of the OLS, IRLS, and NLLS position estimators vs. the proposed 3D CRLB for $N = 50$ sensing elements. This demonstrates that the proposed CRLB is tractable for a massive number of elements and closely aligns with well-known results.

We characterize the positioning performance as a function of GDoP by scaling the ground constellation to meet different integer GDoP values and plotting the CRLB at each value for the same sweep over $\sigma_d$. These results are depicted in Figure 7. While this plot is not particularly interesting, it can directly inform system design decisions; if you have a fixed-ranging precision, you can determine the necessary antenna placement requirements (GDoP) to achieve a certain positioning performance. Likewise, if your antenna placements

are fixed, then you can determine how precise your ranging estimates need to be to achieve a certain positioning performance.

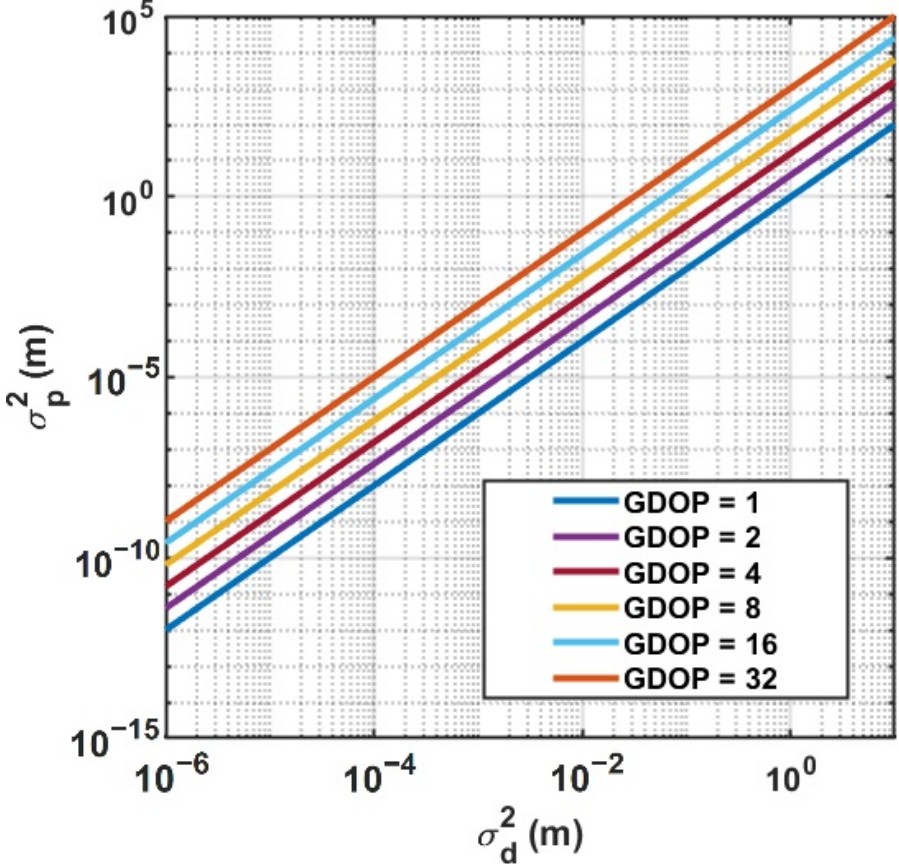

**Figure 7.** The 3D position CRLB vs. ranging precision for several GDoP values. These curves can help inform system design decisions such as antenna placement or required ranging precision. In general, a GDoP of 1 is considered "ideal", between 2 and 10 is considered "good", and anything beyond 20 is considered "poor".

### 5.3. Orientation—3D CRLB

We implemented four orientation estimators in our MATLAB simulation platform: Tri-Axial Attitude Determination (TRIAD) [56]; Davenport's Q-method [57,58]; QUaternion ESTimator (QUEST) [59]; and the optimal linear attitude estimator (OLAE) [60]. We placed two ground nodes $g_1$ and $g_2$ at (0,0,1) and (0,1,0) meters and defined the normalized vector $v_0$ between nodes $G$ and $H$. We then generated normalized antenna position vectors $b_1$ and $b_2$ by rotating $v_k$ using a direction cosine matrix (DCM) with angles $\alpha = \beta = \gamma = \frac{\pi}{6}$. We evaluated $\kappa$ using Equation (53).

In Figure 8, we plot the performance of these estimators versus the 3D orientation CRLB as a function of the positioning precision $\sigma_p$. There is some slight variation between the estimators in different regimes, but they are all consistent with the proposed bound. In Figure 9, we substitute $\kappa$ for $\sigma_p$ following Equation (53) to instead plot the performance as a function of the concentration.

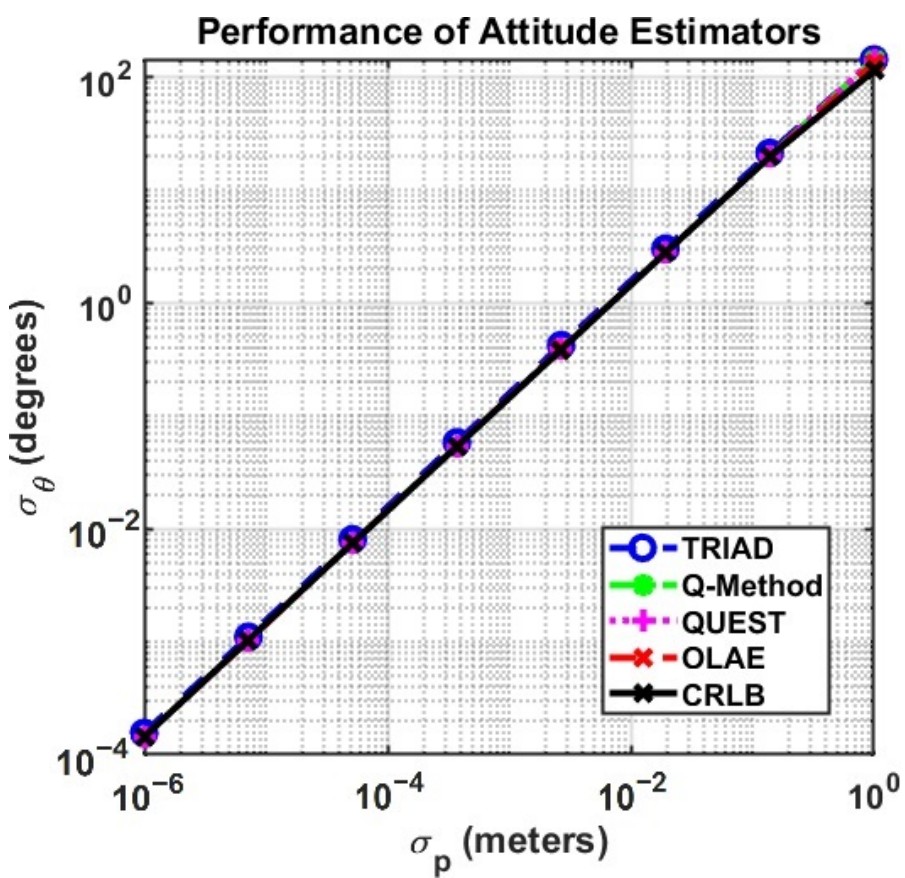

**Figure 8.** The 3D orientation CRLB vs. positioning precision for several attitude estimators.

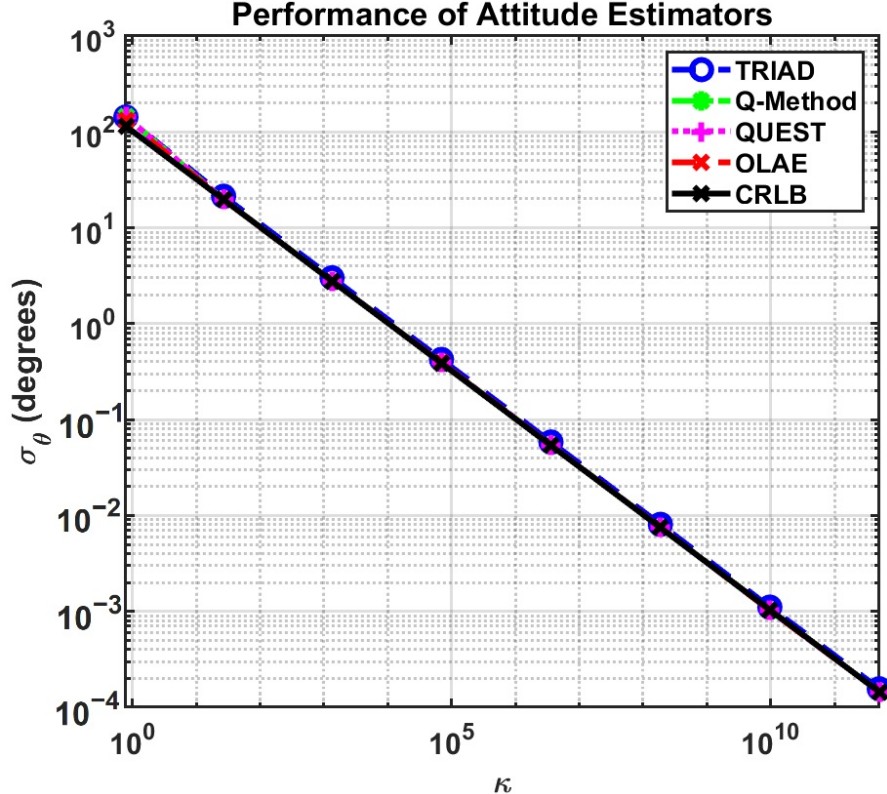

**Figure 9.** The 3D orientation CRLB vs. concentration for several attitude estimators.

## 6. Conclusions

In this manuscript, we derived novel, closed-form, and tractable 2D and 3D CRLBs for position and orientation estimation in a multi-antenna, two-way ranging system. These bounds are specifically formulated as a function of the number of antennas, platform geometry, and geometric dilution of precision. They are also reformulated using different geometric interpretations and plotted as functions of ranging precision and geometric dilution of precision to make these theoretical results more accessible and intuitive to a system design engineer. The performance manifold in Figure 5 also depicts the trade space between ranging precision, positioning precision, and antenna placement, providing a visual reference that does not require any calculations. The proposed CRLBs were compared with several popular position and orientation estimators in a simple MATLAB simulation platform to demonstrate that the bounds are tight, achievable, and consistent with existing work.

This work offers a closed-form and tractable CRLB for position and orientation estimation, but only for the subset of multi-antenna, two-way ranging systems described in Figure 1. This is the natural conclusion to the original motivation, but possible future extensions include a characterization of different antenna geometries similar to the comparisons in [43] and an extension to an N-dimensional formulation that may enable space–time or other high-dimensional processing techniques.

**Author Contributions:** Conceptualization, S.S., A.H., and D.W.B.; data curation, S.S.; formal analysis, S.S.; funding acquisition, D.W.B.; investigation, S.S., S.W., and A.H.; methodology, S.S., A.H., and D.W.B.; project administration, D.W.B.; resources, A.H.; software, S.S., and S.W.; supervision, A.H., and D.W.B.; validation, S.S., and S.W.; visualization, S.S., S.W., and A.H.; writing—original draft preparation, S.S., S.W., and A.H.; Writing—review and editing, S.S., A.H., and D.W.B. All authors have read and agreed to the published version of the manuscript.

**Funding:** This research received no external funding.

**Institutional Review Board Statement:** Not applicable.

**Informed Consent Statement:** Not applicable.

**Data Availability Statement:** The data presented in this study are available on request from the corresponding author. The data are not publicly available due to the policies of Arizona State University regarding pending patents and publications.

**Conflicts of Interest:** The authors declare no conflict of interest.

## Abbreviations

The following abbreviations are used in this manuscript:

| | |
|---|---|
| 2D | Two-dimensional |
| 3D | Three-dimensional |
| CRLB | Cramér–Rao lower bound |
| DCM | Direction cosine matrix |
| FIM | Fisher information matrix |
| GDoP | Geometric dilution of precision |
| IRLS | Iteratively reweighted least squares |
| ISNR | Integrated signal-to-noise ratio |
| LS | Least squares |
| ML | Maximum likelihood |
| NLLS | Non-linear least squares |
| OLAE | Optimal linear attitude estimator |
| OLS | Ordinary least squares |

| | |
|---|---|
| PNT | Positioning, navigation, and timing |
| QUEST | QUaternion ESTimator |
| RF | Radio frequency |
| SWaP-C | Size, weight, power, and cost |
| TDoA | Time difference of arrival |
| ToA | Time of arrival |
| ToA/RSS | Time of arrival/received signal strength |
| ToF | Time of flight |
| TRIAD | Tri-axial attitude determination |
| TWR | Two-way ranging |
| UAM | Urban air mobility |
| UWB | Ultra-wideband |

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
