# Peer review of "Cramér–Rao Lower Bounds on 3D Position and Orientation Estimation in Distributed Ranging Systems"

_applsci, doi:10.3390/app13032008_

Round 1

Reviewer 1 Report

the manuscript reports some useful information and worth to be published.

Author Response

Thank you for your review. We have prepared itemized responses to each suggestion in the attached letter.

Reviewer 2 Report

The paper is relevant to the journal's scope, and can be accepted before the following 3 minor suggestions;

1. Some most recent studies especially in later 2022 if exists may be added. 

2. The whole manuscript may be checked thoroughly for typos, please. 

3. The references provided in this work are in combination, we can't understand what type of work the authors referring to in this study. Please provide a small explanation of the work done in these articles you are citing like, [36-41]. For Example. see lines 74-90 (two paragraphs).

Author Response

(The authors gave the same response as above.)

Reviewer 3 Report

Authors provided a Cramér-Rao Lower Bounds on 3D Position and Orientation Estimation in Distributed Ranging Systems. The present version of the manuscript is not sufficient for publication in Applied Sciences. My suggestion is 'Reject'. I have the following comments.

1. Grammar and spelling errors need to be revised, such as 'positiong estimation' in line 60.

2. The background of Section 2 is unsatisfactory and lacks description and analysis of existing study.

3. The research motivation is not clear, and the innovation is not outstanding.

4. Simulation results are not sufficient to inform the conclusion of 'to inform aerospace system design decisions'.

Author Response

(The authors gave the same response as above.)

Reviewer 4 Report

Necessary citations must be provided for the sentences in the introduction section.

You can have the necessary information about the subject, but the sentences in the introduction must be cited from the relevant places in the literature.

Suggestion: Sections 1 and 2 can be combined. Under the introduction section, the headings can be as follows, respectively.

1. Introduction

1.1. background

1.2. Contributions

The 1.2. section, in which the organization of the paper is defined, is not a preferred writing in articles. It would be appropriate to remove this section.

The novelty of the work must be clearly addressed and discussed, compare your research with existing research findings and highlight novelty, (compare your work with existing research findings and highlight novelty),

Add the following to the continuation of the introduction: 

Gap in the literature 

motivation of study 

Strengths and limitations of the study

For the integrity of the manuscript, match the fonts of your figures with the paper.

There are many personal pronouns used throughout the manuscript. There should be a minimum use of personal pronouns in article writing. In fact, none is correct for writing. (we ,we, we, we) [ use passive voice ]

There is a lot of lumpy reference usage involved.

Suggestion: In addition to the Abbreviations section, the nomenclature section should be added. Greek letters - subscripts and so on

Discuss results in concise and make the way for the future study which need to be addressed.

Conclusion section is missing some perspective related to the future research work.

Author Response

(The authors gave the same response as above.)

Reviewer 5 Report

In this paper, a Cramer-Rao Lower Bounds on 3D Position and Orientation Estimation in Distributed Ranging Systems is proposed. This analysis may be of great interest to readers; the document is well written; however, authors are advised to add future works.

Author Response

(The authors gave the same response as above.)

Round 2

Reviewer 3 Report

  • The author's response is satisfactory and this version can be  accepted.

Reviewer 4 Report

N/A